# AUTOMATA GUIDED SKILL COMPOSITION

## ABSTRACT

Skills learned through reinforcement learning often generalize poorly across tasks and re-training is necessary when presented with a new task. We present a framework that combines techniques in *formal methods* with *reinforcement learning* (RL) that allows for the construction of new skills from existing ones with no additional exploration necessary. Our method also allows for convenient specification of complex temporal dependent tasks using logical expressions. We provide theoretical results for our composition technique and evaluate on a simple grid world simulation as well as a robotic manipulation task.

## 1 INTRODUCTION

Policies learned using reinforcement learning aim to maximize the given reward function and are often difficult to transfer to other problem domains. Skill composition is the process of constructing new skills out of existing ones (policies) with little to no additional learning. In stochastic optimal control, this idea has been adopted by Todorov (2009) and Da Silva et al. (2009) to construct provably optimal control laws based on linearly solvable Markov decision processes.

Temporal logic (TL) is a formal language commonly used in software and digital circuit verification (Baier & Katoen, 2008) as well as formal synthesis (Belta et al., 2017). It allows for convenient expression of complex behaviors and causal relationships. TL has been used by (Tabuada & Pappas, 2004), (Fainekos et al., 2006), (Fainekos et al., 2005) to synthesize provably correct control policies. Aksaray et al. (2016) have also combined TL with Q-learning to learn satisfiable policies in discrete state and action spaces.

We make the distinction between skill composition and multi-task learning/meta-learning where the latter often requires a predefined set of tasks/task distributions to learn and generalize from, whereas the focus of the former is to construct new policies from a library of already learned policies that achieve new tasks (often some combination of the constituent tasks) with little to no additional constraints on task distribution at learning time. In this work, we focus on skill composition with policies learned using automata guided reinforcement learning (Li et al., 2018). We adopt the syntactically co-safe truncated linear temporal logic (scTLTL) as the task specification language. Compared to most heuristic reward structures used in the RL literature, formal specification language has the advantage of semantic rigor and interpretability.

In our framework, skill composition is accomplished by taking the product of finite state automata (FSA). Instead of interpolating/extrapolating among learned skills/latent features, our method is based on graph manipulation of the FSA. Therefore, the outcome is much more transparent. Compared with previous work on skill composition, we impose no constraints on the policy representation or the problem class. We validate our framework in simulation (discrete state and action spaces) and experimentally on a Baxter robot (continuous state and action spaces).

## 2 RELATED WORK

Recent efforts in skill composition have mainly adopted the approach of combining value functions learned using different rewards. Peng et al. (2018) constructs a composite policy by combining the value functions of individual policies using the Boltzmann distribution. With a similar goal, Zhu et al. (2017) achieves task space transfer using deep successor representations (Kulkarni et al., 2016). However, it is required that the reward function be represented as a linear combination of state-action features.

Haarnoja et al. (2018) have showed that when using energy-based models (Haarnoja et al., 2017), an approximately optimal composite policy can result from taking the average of the Q-functions of existing policies. The resulting composite policy achieves the $-AND-$ task composition i.e. the composite policy maximizes the average reward of individual tasks.

van Niekerk et al. (2018) have taken this idea a step further and showed that by combining individual Q-functions using the log-sum-exponential function, the $-OR-$ task composition (the composite policy maximizes the (soft) maximum of the reward of constituent tasks) can be achieved optimally.

We build on the results of (van Niekerk et al., 2018) and show that incorporating temporal logic allows us to compose tasks of greater logical complexity with higher interpretability. Our composite policy is optimal in both $-AND-$ and $-OR-$ task compositions.

## 3 PRELIMINARIES

### 3.1 ENTROPY-REGULARIZED REINFORCEMENT LEARNING AND Q-COMPOSITION

We start with the definition of a Markov Decision Process.

**Definition 1.** *An MDP is defined as a tuple $\mathcal{M} = \langle S, A, p(\cdot|\cdot, \cdot), r(\cdot, \cdot, \cdot)\rangle$, where $S \subseteq \mathbb{R}^n$ is the state space ; $A \subseteq \mathbb{R}^m$ is the action space ($S$ and $A$ can also be discrete sets); $p : S \times A \times S \to [0, 1]$ is the transition function with $p(s'|s, a)$ being the conditional probability density of taking action $a \in A$ at state $s \in S$ and ending up in state $s' \in S$; $r : S \times A \times S \to \mathbb{R}$ is the reward function with $r(s, a, s')$ being the reward obtained by executing action $a$ at state $s$ and transitioning to $s'$.*

In entropy-regularized reinforcement learning (Schulman et al., 2017), the goal is to maximize the following objective

$$J(\pi) = \sum_{t=0}^{T-1} \mathbb{E}^\pi [r_t + \alpha \mathcal{H}(\pi(\cdot|s_t))], \tag{1}$$

where $\pi : S \times A \to [0, 1]$ is a stochastic policy. $\mathbb{E}^\pi$ is the expectation following $\pi$. $\mathcal{H}(\pi(\cdot|s_t))$ is the entropy of $\pi$. $\alpha$ is the temperature parameter. In the limit $\alpha \to 0$, Equation (1) becomes the standard RL objective. The soft Q-learning algorithm introduced by (Haarnoja et al., 2017) optimizes the above objective and finds a policy represented by an energy-based model

$$\pi^\star(a_t|s_t) \propto \exp(-\mathcal{E}(s_t, a_t)) \tag{2}$$

where $\mathcal{E}(s_t, a_t)$ is an energy function that can be represented by a function approximator.

Let $r_t^{ent} = r_t + \alpha \mathcal{H}(\pi(\cdot|s_t))$, the state-action value function (Q-function) following $\pi$ is defined as $Q^\pi(s, a) = \mathbb{E}^\pi[\sum_{t=0}^{T-1} r_t^{ent}|s_0 = s, a_0 = a]$. Suppose we have a set of $n$ tasks indexed by $i, i \in \{0, ..., n\}$, each task is defined by an MDP $\mathcal{M}_i$ that differs only in their reward function $r_i$. Let $Q_\alpha^{\pi_i}$ be the optimal entropy-regularized Q-function. Authors of (van Niekerk et al., 2018) provide the following results

**Theorem 1.** *Define vectors $\boldsymbol{r} = [r_1, ..., r_n]$, $\boldsymbol{Q}_\alpha^{\pi^\star} = [Q_\alpha^{\pi_1^\star}, ..., Q_\alpha^{\pi_n^\star}]$. Given a set of non-negative weights $\boldsymbol{w}$ with $||\boldsymbol{w}|| = 1$, the optimal Q-function for a new task defined by $r = \alpha \log(|| \exp(\boldsymbol{r}/\alpha)||_{\boldsymbol{w}})$ is given by*

$$Q_\alpha^{\pi^\star} = \alpha \log(|| \exp(\boldsymbol{Q}_\alpha^{\pi^\star}/\alpha)||_{\boldsymbol{w}}), \tag{3}$$

*where $|| \cdot ||_{\boldsymbol{w}}$ is the weighted 1-norm.*

The authors proceed to provide the following corollary

**Corollary 1.** $\max \boldsymbol{Q}_\alpha^{\pi^\star} \uparrow Q_0^{\pi^\star}$ *as $\alpha \to 0$, where $Q_0^{\pi^\star}$ is the optimal Q-function for the objective $J(\pi) = \sum_{t=0}^{T-1} \mathbb{E}^\pi[r_t]$.*

Corollary 1 states that in the low temperature limit, the maximum of the optimal entropy-regularized Q-functions approaches the standard optimal Q-function

### 3.2 scTLTL and Finite State Automata

We consider tasks specified with *syntactically co-safe Truncated Linear Temporal Logic* (scTLTL) which is derived from truncated linear temporal logic(TLTL) (Li et al., 2018). The syntax of scTLTL is defined as

$$\phi := \top \mid f(s) < c \mid \neg\phi \mid \phi \wedge \psi \mid \Diamond\phi \mid \phi\, \mathcal{U}\, \psi \mid \phi\, \mathcal{T}\, \psi \mid \bigcirc \phi \tag{4}$$

where $\top$ is the True Boolean constant. $s \in S$ is a MDP state in Definition 1. $f(s) < c$ is a predicate over the MDP states where $c \in \mathbb{R}$. $\neg$ (negation/not), $\wedge$ (conjunction/and) are Boolean connectives. $\Diamond$ (eventually), $\mathcal{U}$ (until), $\mathcal{T}$ (then), $\bigcirc$ (next), are temporal operators. $\Rightarrow$ (implication) and and $\vee$ (disjunction/or) can be derived from the above operators.

We denote $s_t \in S$ to be the MDP state at time $t$, and $s_{t:t+k}$ to be a sequence of states (state trajectory) from time $t$ to $t + k$, i.e., $s_{t:t+k} = s_t s_{t+1}...s_{t+k}$. The Boolean semantics of scTLTL is defined as:

$$
\begin{aligned}
s_{t:t+k} &\models f(s) < c & \Leftrightarrow\quad & f(s_t) < c, \\
s_{t:t+k} &\models \neg\phi & \Leftrightarrow\quad & \neg(s_{t:t+k} \models \phi), \\
s_{t:t+k} &\models \phi \Rightarrow \psi & \Leftrightarrow\quad & (s_{t:t+k} \models \phi) \Rightarrow (s_{t:t+k} \models \psi), \\
s_{t:t+k} &\models \phi \wedge \psi & \Leftrightarrow\quad & (s_{t:t+k} \models \phi) \wedge (s_{t:t+k} \models \psi), \\
s_{t:t+k} &\models \phi \vee \psi & \Leftrightarrow\quad & (s_{t:t+k} \models \phi) \vee (s_{t:t+k} \models \psi), \\
s_{t:t+k} &\models \bigcirc\phi & \Leftrightarrow\quad & (s_{t+1:t+k} \models \phi) \wedge (k > 0), \\
s_{t:t+k} &\models \Diamond\phi & \Leftrightarrow\quad & \exists t' \in [t, t+k)\; s_{t':t+k} \models \phi, \\
s_{t:t+k} &\models \phi\, \mathcal{U}\, \psi & \Leftrightarrow\quad & \exists t' \in [t, t+k)\; \text{s.t.}\; s_{t':t+k} \models \psi \\
& & & \wedge\, (\forall t'' \in [t, t')\; s_{t'':t'} \models \phi), \\
s_{t:t+k} &\models \phi\, \mathcal{T}\, \psi & \Leftrightarrow\quad & \exists t' \in [t, t+k)\; \text{s.t.}\; s_{t':t+k} \models \psi \\
& & & \wedge\, (\exists t'' \in [t, t')\; s_{t'':t'} \models \phi).
\end{aligned}
$$

A trajectory $s_{0:T}$ is said to satisfy formula $\phi$ if $s_{0:T} \models \phi$.

The quantitative semantics (also referred to as robustness) is defined recursively as

$$
\begin{aligned}
\rho(s_{t:t+k}, \top) &= \rho_{max}, \\
\rho(s_{t:t+k}, f(s_t) < c) &= c - f(s_t), \\
\rho(s_{t:t+k}, \neg\phi) &= -\rho(s_{t:t+k}, \phi), \\
\rho(s_{t:t+k}, \phi \Rightarrow \psi) &= \max(-\rho(s_{t:t+k}, \phi), \rho(s_{t:t+k}, \psi)) \\
\rho(s_{t:t+k}, \phi_1 \wedge \phi_2) &= \min(\rho(s_{t:t+k}, \phi_1), \rho(s_{t:t+k}, \phi_2)), \\
\rho(s_{t:t+k}, \phi_1 \vee \phi_2) &= \max(\rho(s_{t:t+k}, \phi_1), \rho(s_{t:t+k}, \phi_2)), \\
\rho(s_{t:t+k}, \bigcirc\phi) &= \rho(s_{t+1:t+k}, \phi)\; (k > 0), \\
\rho(s_{t:t+k}, \Diamond\phi) &= \max_{t' \in [t, t+k)} (\rho(s_{t':t+k}, \phi)), \\
\rho(s_{t:t+k}, \phi\, \mathcal{U}\, \psi) &= \max_{t' \in [t, t+k)} (\min(\rho(s_{t':t+k}, \psi), \\
& \qquad \min_{t'' \in [t, t')} \rho(s_{t'':t'}, \phi))), \\
\rho(s_{t:t+k}, \phi\, \mathcal{T}\, \psi) &= \max_{t' \in [t, t+k)} (\min(\rho(s_{t':t+k}, \psi), \\
& \qquad \max_{t'' \in [t, t')} \rho(s_{t'':t'}, \phi))),
\end{aligned}
$$

where $\rho_{max}$ represents the maximum robustness value. A robustness of greater than zero implies that $s_{t:t+k}$ satisfies $\phi$ and vice versa ($\rho(s_{t:t+k}, \phi) > 0 \Rightarrow s_{t:t+k} \models \phi$ and $\rho(s_{t:t+k}, \phi) < 0 \Rightarrow s_{t:t+k} \not\models \phi$). The robustness is used as a measure of the level of satisfaction of a trajectory $s_{0:T}$ with respect to a scTLTL formula $\phi$.

**Definition 2.** *An FSA corresponding to a scTLTL formula $\phi$. is defined as a tuple $\mathcal{A}_\phi = \langle \mathbb{Q}_\phi, \Psi_\phi, q_{\phi,0}, p_\phi(\cdot|\cdot), \mathcal{F}_\phi \rangle$, where $\mathbb{Q}_\phi$ is a set of automaton states; $\Psi_\phi$ is the input alphabet (a set of first order logic formula); $q_{\phi,0} \in \mathbb{Q}_\phi$ is the initial state; $p_\phi : \mathbb{Q}_\phi \times \mathbb{Q}_\phi \to [0,1]$ is a conditional probability defined as*

$$p_\phi(q_{\phi,j}|q_{\phi,i}) = \begin{cases} 1 & \psi_{q_{\phi,i},q_{\phi,j}} \text{ is true} \\ 0 & \text{otherwise.} \end{cases}$$

$$or \tag{5}$$

$$p_\phi(q_{\phi,j}|q_{\phi,i},s) = \begin{cases} 1 & \rho(s, \psi_{q_{\phi,i},q_{\phi,j}}) > 0 \\ 0 & \text{otherwise.} \end{cases}$$

*$\mathcal{F}_\phi$ is a set of final automaton states.*

Here $q_{\phi,i}$ is the $i^{th}$ automaton state of $\mathcal{A}_\phi$. $\psi_{q_{\phi,i},q_{\phi,j}} \in \Psi_\phi$ is the predicate guarding the transition from $q_{\phi,i}$ to $q_{\phi,j}$. Because $\psi_{q_{\phi,i},q_{\phi,j}}$ is a predicate without temporal operators, the robustness $\rho(s_{t:t+k}, \psi_{q_{\phi,i},q_{\phi,j}})$ is only evaluated at $s_t$. Therefore, we use the shorthand $\rho(s_t, \psi_{q_{\phi,i},q_{\phi,j}}) = \rho(s_{t:t+k}, \psi_{q_{\phi,i},q_{\phi,j}})$. The translation from a TLTL formula to a FSA can be done automatically with available packages like Lomap (Vasile, 2017).

### 3.3 FSA Augmented MDP

The FSA Augmented MDP is defined as follows

**Definition 3.** *(Li et al., 2018) An FSA augmented MDP corresponding to scTLTL formula $\phi$ (constructed from FSA $\langle \mathbb{Q}_\phi, \Psi_\phi, q_{\phi,0}, p_\phi(\cdot|\cdot), \mathcal{F}_\phi \rangle$ and MDP $\langle S, A, p(\cdot|\cdot,\cdot), r(\cdot,\cdot,\cdot) \rangle$) is defined as $\mathcal{M}_\phi = \langle \tilde{S}, A, \tilde{p}(\cdot|\cdot,\cdot), \tilde{r}(\cdot,\cdot), \mathcal{F}_\phi \rangle$ where $\tilde{S} \subseteq S \times \mathbb{Q}_\phi$, $\tilde{p}(\tilde{s}'|\tilde{s},a)$ is the probability of transitioning to $\tilde{s}'$ given $\tilde{s}$ and $a$,*

$$\tilde{p}(\tilde{s}'|\tilde{s},a) = p\big((s',q'_\phi)|(s,q_\phi),a\big) = \begin{cases} p(s'|s,a) & p_\phi(q\phi'|q_\phi,s) = 1 \\ 0 & \text{otherwise.} \end{cases} \tag{6}$$

*$p_\phi$ is defined in Equation (5). $\tilde{r} : \tilde{S} \times \tilde{S} \to \mathbb{R}$ is the FSA augmented reward function, defined by*

$$\tilde{r}(\tilde{s}, \tilde{s}') = \rho(s', D_\phi^{q_\phi}), \tag{7}$$

*where $D_\phi^{q_\phi} = \bigvee_{q'_\phi \in \Omega_{q_\phi}} \psi_{q_\phi,q'_\phi}$ represents the disjunction of all predicates guarding the transitions that originate from $q_\phi$ ($\Omega_{q_\phi}$ is the set of automata states that are connected with $q$ through outgoing edges).*

A policy $\pi_\phi^\star$ is said to satisfy $\phi$ if

$$\pi_\phi^\star = \arg\max_{\pi_\phi} \mathbb{E}^{\pi_\phi}[\mathbb{1}(\rho(s_{0:T}, \phi) > 0)]. \tag{8}$$

where $\mathbb{1}(\rho(s_{0:T}, \phi) > 0)$ is an indicator function with value 1 if $\rho(s_{0:T}, \phi) > 0$ and 0 otherwise. As is mentioned in the original paper, there can be multiple policies that meet the requirement of Equation (8), therefore, a discount factor is used to find a maximally satisfying policy - one that leads to satisfaction in the least amount of time.

The FSA augmented MDP $\mathcal{M}_\phi$ establishes a connection between the TL specification and the standard reinforcement learning problem. A policy learned using $\mathcal{M}_\phi$ has implicit knowledge of the FSA through the automaton state $q_\phi \in \mathcal{Q}_\phi$. We will take advantage of this characteristic in our skill composition framework.

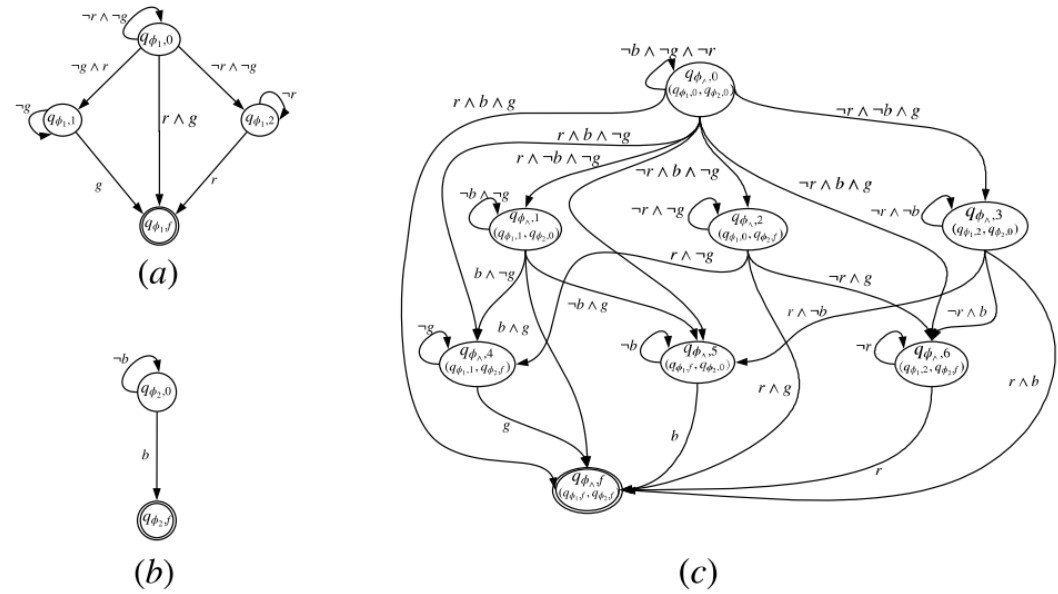

**Figure 1 :** FSA for **(a)** $\phi_1 = \Diamond a \wedge \Diamond b$. **(b)** $\phi_2 = \Diamond c$. **(c)** $\phi_\wedge = \phi_1 \wedge \phi_2$.

## 4 PROBLEM FORMULATION

**Problem 1.** *Given two scTLTL formula $\phi_1$ and $\phi_2$ and their optimal Q-functions $Q^\star_{\phi_1}$ and $Q^\star_{\phi_2}$, obtain the optimal policy $\pi^\star_{\phi_\wedge}$ that satisfies $\phi_\wedge = \phi_1 \wedge \phi_2$ and $\pi^\star_{\phi_\vee}$ that satisfies $\phi_\vee = \phi_1 \vee \phi_2$.*

Here $Q^\star_{\phi_1}$ and $Q^\star_{\phi_2}$ can be the optimal Q-functions for the entropy-regularized MDP or the standard MDP. Problem 1 defines the problem of skill composition: given two policies each satisfying a scTLTL specification, construct the policy that satisfies the conjunction $(-AND-)$/disjunction $(-OR-)$ of the given specifications. Solving this problem is useful when we want to break a complex task into simple and manageable components, learn a policy that satisfies each component and "stitch" all the components together so that the original task is satisfied. It can also be the case that as the scope of the task grows with time, the original task specification is amended with new items. Instead of having to re-learn the task from scratch, we can learn only policies that satisfies the new items and combine them with the old policy.

## 5 AUTOMATA GUIDED SKILL COMPOSITION

In this section, we provide a solution for Problem 1 by constructing the FSA of $\phi_\wedge$ from that of $\phi_1$ and $\phi_2$ and using $\phi_\wedge$ to synthesize the policy for the combined skill. We start with the following definition.

**Definition 4.** *Given $\mathcal{A}_{\phi_1} = \langle \mathbb{Q}_{\phi_1}, \Psi_{\phi_1}, q_{\phi_1,0}, p_{\phi_1}, \mathcal{F}_{\phi_1} \rangle$ and $\mathcal{A}_{\phi_2} = \langle \mathbb{Q}_{\phi_2}, \Psi_{\phi_2}, q_{\phi_2,0}, p_{\phi_2}, \mathcal{F}_{\phi_2} \rangle$ corresponding to formulas $\phi_1$ and $\phi_2$, the FSA of $\phi_\wedge = \phi_1 \wedge \phi_2$ is the product automaton of $\mathcal{A}_{\phi_1}$ and $\mathcal{A}_{\phi_1}$, i.e. $\mathcal{A}_{\phi_\wedge} = \phi_1 \wedge \phi_2 = \mathcal{A}_{\phi_1} \times \mathcal{A}_{\phi_2} = \langle \mathbb{Q}_{\phi_\wedge}, \Psi_{\phi_\wedge}, q_{\phi_\wedge,0}, p_{\phi_\wedge}, \mathcal{F}_{\phi_\wedge} \rangle$ where $\mathbb{Q}_{\phi_\wedge} \subseteq \mathbb{Q}_{\phi_1} \times \mathbb{Q}_{\phi_2}$ is the set of product automaton states, $q_{\phi_\wedge,0} = (q_{\phi_1,0}, q_{\phi_2,0})$ is the product initial state, $\mathcal{F}_{\phi_\wedge} \subseteq \mathcal{F}_{\phi_1} \cap \mathcal{F}_{\phi_2}$ are the final accepting states. Following Definition 2, for states $q_{\phi_\wedge} = (q_{\phi_1}, q_{\phi_2}) \in \mathbb{Q}_{\phi_\wedge}$ and $q'_{\phi_\wedge} = (q'_{\phi_1}, q'_{\phi_2}) \in \mathbb{Q}_{\phi_\wedge}$, the transition probability $p_{\phi_\wedge}$ is defined as*

$$p_{\phi_\wedge}(q'_{\phi_\wedge}|q_{\phi_\wedge}) = \begin{cases} 1 & p_{\phi_1}(q'_{\phi_1}|q_{\phi_1})p_{\phi_2}(q'_{\phi_2}|q_{\phi_2}) = 1 \\ 0 & otherwise. \end{cases} \tag{9}$$

**Example 1.** *Figure 1 illustrates the FSA of $\mathcal{A}_{\phi_1}$ and $\mathcal{A}_{\phi_2}$ and their product automaton $\mathcal{A}_{\phi_\wedge}$. Here $\phi_1 = \Diamond r \wedge \Diamond g$ which entails that both $r$ and $g$ needs to be true at least once (order does not matter), and $\phi_2 = \Diamond b$. The resultant product corresponds to the formula $\phi = \Diamond r \wedge \Diamond g \wedge \Diamond b$.*

We provide the following theorem on automata guided skill composition

**Theorem 2.** *Let $\boldsymbol{Q}_\phi^{\pi^\star} = [Q_{\phi_1}^{\pi_1^\star}, ..., Q_{\phi_n}^{\pi_n^\star}]$ be a vector with entries $Q_{\phi_i}^{\pi_i^\star}$ being the optimal Q-function for the FSA augmented MDP $\mathcal{M}_{\phi_i}$. The optimal Q-function for $\mathcal{M}_{\phi_\wedge}$ where $\phi_\wedge = \bigwedge_i \phi_i$ is $Q_{\phi_\wedge}^{\pi_\wedge^\star} = \max(\boldsymbol{Q}_\phi^{\pi^\star})$.*

*Proof.* For $q_{\phi_\wedge} = (q_{\phi_1}, q_{\phi_2}) \in \mathbb{Q}_{\phi_\wedge}$, let $\Psi_{q_{\phi_\wedge}}$, $\Psi_{q_{\phi_1}}$ and $\Psi_{q_{\phi_2}}$ denote the set of predicates guarding the edges originating from $q_{\phi_\wedge}$, $q_{\phi_1}$ and $q_{\phi_2}$ respectively. Equation (9) entails that a transition at $q_{\phi_\wedge}$ in the product automaton $\mathcal{A}_{\phi_\wedge}$ exists only if corresponding transitions at $q_{\phi_1}$, $q_{\phi_2}$ exist in $\mathcal{A}_{\phi_1}$ and $\mathcal{A}_{\phi_2}$ respectively. Therefore, $\psi_{q_{\phi_\wedge}, q'_{\phi_\wedge}} = \psi_{q_{\phi_1}, q'_{\phi_1}} \wedge \psi_{q_{\phi_2}, q'_{\phi_2}}$, for $\psi_{q_{\phi_\wedge}, q'_{\phi_\wedge}} \in \Psi_{q_{\phi_\wedge}}, \psi_{q_{\phi_1}, q'_{\phi_1}} \in \Psi_{q_{\phi_1}}, \psi_{q_{\phi_2}, q'_{\phi_2}} \in \Psi_{q_{\phi_2}}$ (here $q'_{\phi_i}$ is a state such that $p_{\phi_i}(q'_{\phi_i} | q_{\phi_i}) = 1$). Therefore, we have

$$D_{\phi_\wedge}^{q_{\phi_\wedge}} = \bigvee_{q'_{\phi_1}, q'_{\phi_2}} (\psi_{q_{\phi_1}, q'_{\phi_1}} \wedge \psi_{q_{\phi_2}, q'_{\phi_2}}) \tag{10}$$

where $q'_{\phi_1}, q'_{\phi_2}$ don't equal to $q_{\phi_1}, q_{\phi_2}$ at the same time (to avoid self looping edges). Using the fact that $\psi_{q_{\phi_i}, q_{\phi_i}} = \neg \bigvee_{q'_{\phi_i} \neq q_{\phi_i}} \psi_{q_{\phi_i}, q'_{\phi_i}}$ and repeatedly applying the distributive laws $(\Delta \wedge \Omega_1) \vee (\Delta \wedge \Omega_2) = \Delta \wedge (\Omega_1 \vee \Omega_2)$ and $(\Delta \vee \Omega_1) \wedge (\Delta \vee \Omega_2) = \Delta \vee (\Omega_1 \wedge \Omega_2)$ to $D_{\phi_\wedge}^{q_{\phi_\wedge}}$, we arrive at

$$D_{\phi_\wedge}^{q_{\phi_\wedge}} = \Big( \bigvee_{q'_{\phi_1} \neq q_{\phi_1}} \psi_{q_{\phi_1}, q'_{\phi_1}} \Big) \vee \Big( \bigvee_{q'_{\phi_2} \neq q_{\phi_2}} \psi_{q_{\phi_2}, q'_{\phi_2}} \Big) = D_{\phi_1}^{q_{\phi_1}} \vee D_{\phi_2}^{q_{\phi_2}}. \tag{11}$$

Let $\tilde{r}_{\phi_\wedge}, \tilde{r}_{\phi_1}, \tilde{r}_{\phi_2}$ and $\tilde{s}_{\phi_\wedge}, \tilde{s}_{\phi_1}, \tilde{s}_{\phi_2}$ be the reward functions and states for FSA augmented MDP $\mathcal{M}_{\phi_\wedge}, \mathcal{M}_{\phi_1}, \mathcal{M}_{\phi_2}$ respectively. $s_{\phi_\wedge}, s_{\phi_1}, s_{\phi_2}$ are the states for the corresponding MDPs. Plugging Equation (11) into Equation (7) and using the robustness definition for disjunction results in

$$\begin{aligned} \tilde{r}_{\phi_\wedge}(\tilde{s}_{\phi_\wedge}, \tilde{s}'_{\phi_\wedge}) &= \rho(s'_{\phi_\wedge}, D_{\phi_\wedge}^{q_{\phi_\wedge}}) \\ &= \rho(s'_{\phi_\wedge}, D_{\phi_1}^{q_{\phi_1}} \vee D_{\phi_2}^{q_{\phi_2}}) \\ &= \max\big(\rho(s'_{\phi_1}, D_{\phi_1}^{q_{\phi_1}}), \rho(s'_{\phi_2}, D_{\phi_2}^{q_{\phi_2}})\big) \\ &= \max\big(\tilde{r}_{\phi_1}(\tilde{s}_{\phi_1}, \tilde{s}'_{\phi_1}), \tilde{r}_{\phi_2}(\tilde{s}_{\phi_2}, \tilde{s}'_{\phi_2})\big). \end{aligned} \tag{12}$$

Looking at Theorem 1, the log-sum-exp of the composite reward $\mathbf{r} = \alpha \log(|| \exp(\boldsymbol{r}/\alpha) ||_{\boldsymbol{w}})$ is in fact an approximation of the maximum function. In the low temperature limit we have $r \to \max(\boldsymbol{r})$ as $\alpha \to 0$. Applying Corollary 1 results in Theorem 2. $\square$

Having obtained the optimal Q-function, a policy can be constructed by taking the greedy step with respective to the Q-function in the discrete action case. For the case of continuous action space where the policy is represented by a function approximator, the policy update procedure in actor-critic methods can be used to extract a policy from the Q-function.

In our framework, $-AND-$ and $-OR-$ task compositions follow the same procedure (Theorem 2). The only difference is the termination condition. For $-AND-$ task, the final state $\mathcal{F}_{\phi_\wedge} = \bigcap \mathcal{F}_{\phi_i}$ in Definition 4 needs to be reached (i.e. all the constituent FSAs are required to reach terminal state, as in state $q_{\phi_\wedge, f}$ in Figure 1 ). Whereas for the $-OR-$ task, only $\mathcal{F}_{\phi_\vee} = \bigcup \mathcal{F}_{\phi_i}$ needs to be reached (one of states $q_{\phi_\wedge, 2}, q_{\phi_\wedge, 4}, q_{\phi_\wedge, 5}, q_{\phi_\wedge, 6}, q_{\phi_\wedge, f}$ in Figure 1 ). A summary of the composition procedure is provided in Algorithm 1.

In Algorithm 1, steps 3 and 4 seeks to obtain the optimal policy and Q-function using any off-policy actor critic RL algorithm. $\mathcal{B}_{\phi_{1,2}}$ are the replay buffers collected while training for each skill. Step

---

**Algorithm 1** Automata Guided Skill Composition

---

1: **Inputs**: scTLTL task specification $\phi_1$ and $\phi_2$, randomly initialized policies $\pi_{\phi_1}$, $\pi_{\phi_2}$ and action-value functions $Q_{\phi_1}^{\pi_{\phi_1}}$, $Q_{\phi_2}^{\pi_{\phi_2}}$. State and action spaces of the MDP.
2: Construct FSA augmented MDPs $\mathcal{M}_{\phi_1}$ and $\mathcal{M}_{\phi_2}$          $\triangleright$ using Definition 3
3: $\pi_{\phi_1}^\star, Q_{\phi_1}^{\pi_{\phi_1}^\star}, \mathcal{B}_{\phi_1} \leftarrow ActorCritic(\mathcal{M}_{\phi_1})$        $\triangleright$ learns the optimal policy and Q-function
4: $\pi_{\phi_2}^\star, Q_{\phi_2}^{\pi_{\phi_2}^\star}, \mathcal{B}_{\phi_2} \leftarrow ActorCritic(\mathcal{M}_{\phi_2})$
5: $Q_{\phi_2 \wedge \phi_2}^{\pi_{\phi_2 \wedge \phi_2}^\star} = \max(Q_{\phi_1}^{\pi_{\phi_1}^\star}, Q_{\phi_2}^{\pi_{\phi_2}^\star})$   $\triangleright$ construct the optimal composed Q-function using Theorem 2
6: $\mathcal{B}_{\phi_\wedge} \leftarrow ConstructProductBuffer(\mathcal{B}_{\phi_1}, \mathcal{B}_{\phi_2})$
7: $\pi_{\phi_2 \wedge \phi_2}^\star \leftarrow ExtractPolicy(Q_{\phi_2 \wedge \phi_2}^{\pi_{\phi_2 \wedge \phi_2}^\star}, \mathcal{B}_{\phi_\wedge})$
8: **return** $\pi_{\phi_2 \wedge \phi_2}^\star$

---

6 constructs the product replay buffer for policy extraction. This step is necessary because each $\mathcal{B}_{\phi_i}$ contains state of form $(s, q_i)$, $i \in \{1, 2\}$ whereas the composed policy takes state $(s, q_1, q_2)$ as input (as in Definition 4). Therefore, we transform each experience $((s, q_i), a, (s', q_i'), r)$ to $((s, q_i, q_{j \neq i}), a, (s', q_i', q_{j \neq i}'), r)$ where $q_{j \neq i}$ is chosen at random from the automaton states of $\mathcal{A}_{\phi_j}$ and $q_{j \neq i}'$ is calculated from Equation (6). The reward $r$ will not be used in policy extraction as the Q-function will not be updated. Step 7 extracts the optimal composed policy from the optimal composed Q-function (this corresponds to running only the policy update step in the actor critic algorithm).

## 6    CASE STUDIES

We evaluate the our composition method in two environments. The first is a simple 2D grid world environment that is used as for proof of concept and policy visualization. The second is a robot manipulation environment.

### 6.1    GRID WORLD

Consider an agent that navigates in a $8 \times 10$ grid world. Its MDP state space is $S : X \times Y$ where $x, y \in X, Y$ are its integer coordinates on the grid. The action space is $A : [up, down, left, right, stay]$. The transition is such that for each action command, the agent follows that command with probability 0.8 or chooses a random action with probability 0.2. We train the agent on two tasks, $\phi_1 = \Diamond r \wedge \Diamond g$ and $\phi_2 = \Diamond b$ (same as in Example 1). The regions are defined by the predicates $r = (1 < x < 3) \wedge (1 < y < 3)$ and $g = (4 < x < 6) \wedge (4 < y < 6)$. Because the coordinates are integers, $a$ and $b$ define a point goal rather than regions. $\phi_2$ expresses a similar task for $b = (1 < x < 3) \wedge (6 < y < 8)$. Figure 1 shows the FSA for each task.

We apply standard tabular Q-learning Watkins (1989) on the FSA augmented MDP of this environment. For all experiments, we use a discount factor of 0.95, learning rate of 0.1, episode horizon of 200 steps, a random exploration policy and a total number of 2000 update steps which is enough to reach convergence (learning curve is not presented here as it is not the focus of this paper).

Figure 2 (a) and (b) show the learned optimal policies extracted by $\pi_{\phi_i}^\star(x, y, q_{\phi_i}) = \arg\max_a Q_{\phi_i}^\star(x, y, q_{\phi_i}, a)$. We plot $\pi_{\phi_i}^\star(x, y, q_{\phi_i})$ for each $q_{\phi_i}$ and observe that each represents a sub-policy whose goal is given by Equation 7.

Figure 2 (c) shows the composed policy of $\phi_\wedge = \phi_1 \wedge \phi_2$ using Theorem 2. It can be observed that the composed policy is able to act optimally in terms maximizing the expected sum of discounted rewards given by Equation (12). Following the composed policy and transitioning the FSA in Figure 1 (b) will in fact satisfy $\phi_\wedge$ ($-AND-$). As discussed in the previous section, if the $-OR-$ task is desired, following the same composed policy and terminate at any of the states $q_{\phi_\wedge, 2}$, $q_{\phi_\wedge, 4}$, $q_{\phi_\wedge, 5}$, $q_{\phi_\wedge, 6}$, $q_{\phi_\wedge, f}$ will satisfy $\phi_\vee = \phi_1 \vee \phi_2$.

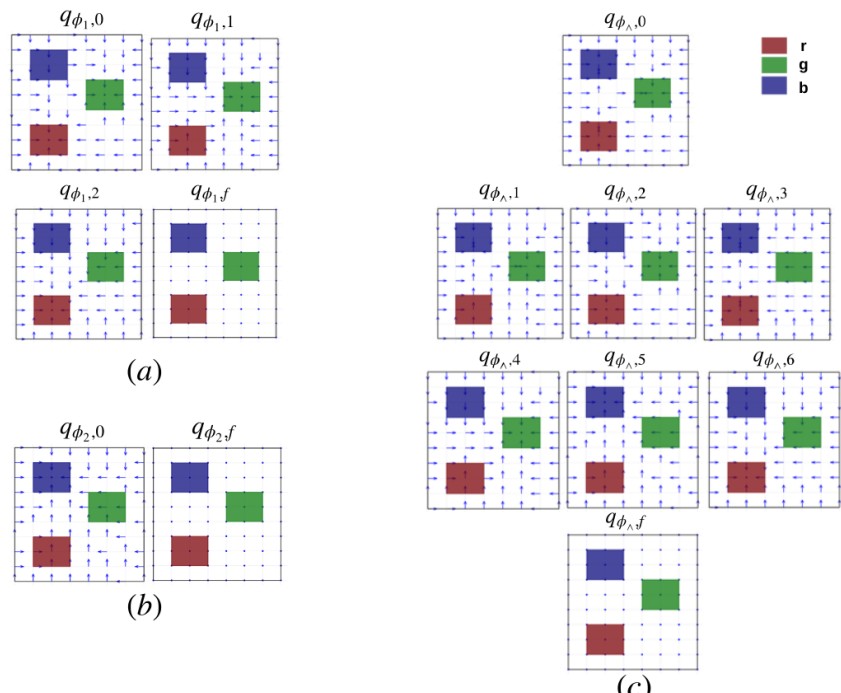

**Figure 2 :** Policies for **(a)** $\phi_1 = \Diamond a \wedge \Diamond b$. **(b)** $\phi_2 = \Diamond c$. **(c)** $\phi_\wedge = \phi_1 \wedge \phi_2$. The agent moves in a $8 \times 10$ gridworld with 3 labeled regions. The agent has actions [*up*, *down*, *left*, *right*, *stay*] where the directional actions are represented by arrows, *stay* is represented by the blue dot.

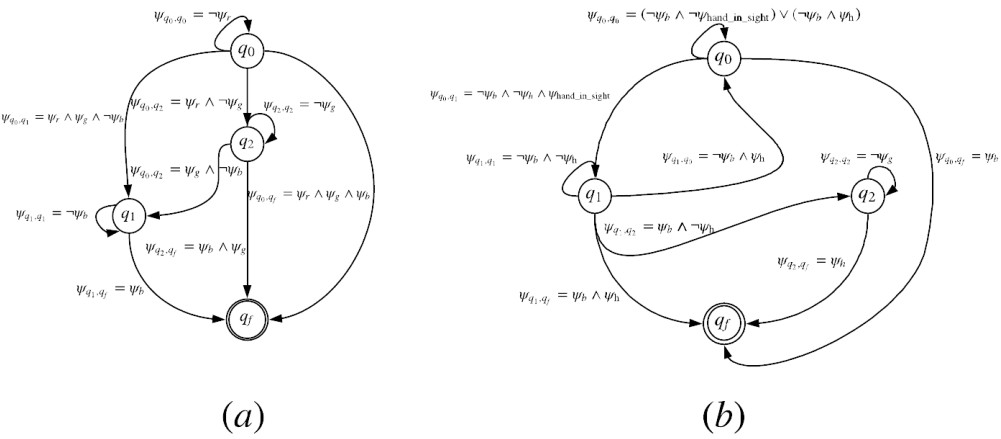

**Figure 3 :** The FSA for **(a)**:$\phi_{traverse} = \Diamond(\psi_r \wedge \Diamond(\psi_g \wedge \Diamond\psi_b))$ **b** $\phi_{interrupt} = (\psi_{hand\_in\_sight} \Rightarrow \Diamond\psi_h) \; \mathcal{U} \; \psi_b$. The subscripts $\phi_1$ and $\phi_2$ are dropped for clarity.

## 6.2 ROBOTIC MANIPULATION

In this sub-section, we test our method on a more complex manipulation task.

### 6.2.1 EXPERIMENT SETUP

Figure 4 (a) presents our experiment setup. Our policy controls the 7 degree-of-freedom joint velocities of the right arm of a Baxter robot. In front of the robot are three circular regions (red, green,

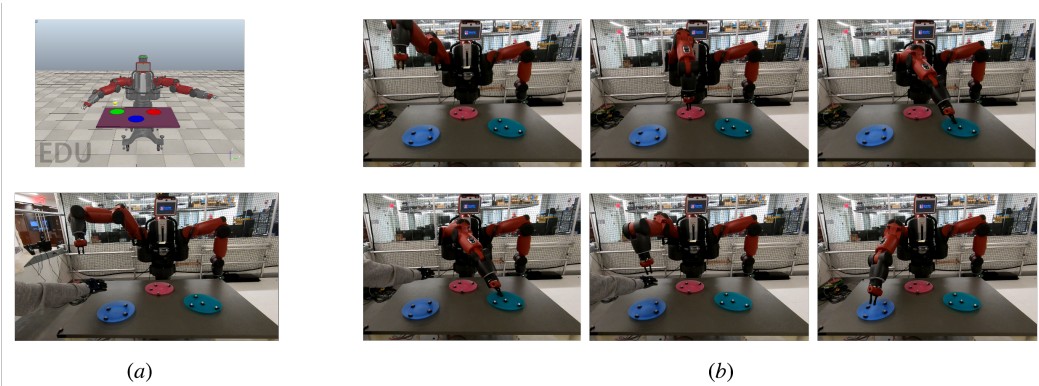

$(a)$ $(b)$

**Figure 4 :** **(a)**: The upper figure shows our simulation environment in V-REP Rohmer et al. (2013) and the lower shows the corresponding experimental environment. **(b)** An execution trace of policy $\pi_{\phi_\wedge}$ where $\phi_\wedge = \phi_{traverse} \wedge \phi_{interrupt}$.

blue plates) and it has to learn to traverse in user specified ways. The positions of the plates are tracked by motion capture systems and thus fully observable. In addition, we also track the position of one of the user's hands (by wearing a glove with trackers attached). Our MDP state space is 22 dimensional that includes 7 joint angles, xyz positions of the three regions (denoted by $\boldsymbol{p}^r, \boldsymbol{p}^g, \boldsymbol{p}^b$), the user's hand ($\boldsymbol{p}^h$) and the robot's end-effector ($\boldsymbol{p}^{ee}$). State and action spaces are continuous in this case.

We define the following predicates

1. $\psi_i = ||\boldsymbol{p}^i - \boldsymbol{p}^{ee}|| < \epsilon,\ i \in \{r, g, b, h\}$ where $\epsilon$ is a threshold which we set to be 5 centimeters. $\psi_i$ constrains the relative distance between the robot's end-effector and the selected object.

2. $\psi_{hand\_in\_sight} = (x_{min} < \boldsymbol{p}_x^h < x_{max}) \wedge (y_{min} < \boldsymbol{p}_y^h < y_{max}) \wedge (z_{min} < \boldsymbol{p}_z^h < z_{max})$. This predicate evaluates to true if the user's hand appears in the cubic region defined by $[x_{min}, x_{max}, y_{min}, y_{max}, z_{min}, z_{max}]$. In this experiment, we take this region to be 40 centimeters above the table (length and width the same as the table).

We test our method on the following composition task

1. $\phi_{traverse} = \Diamond(\psi_r \wedge \Diamond(\psi_g \wedge \Diamond\psi_b))$
   Description: traverse the three regions in the order of red, green, blue.

2. $\phi_{interrupt} = (\psi_{hand\_in\_sight} \Rightarrow \Diamond\psi_h)\ \mathcal{U}\ \psi_b$
   Description: before reaching the blue region, if the user's hand appears in sight, then eventually reach for the user's hand and the blue region, otherwise just reach for the blue region.

3. $\phi_\wedge = \phi_{traverse} \wedge \phi_{interrupt}$
   Description: conjunction of the first two tasks.

The FSAs for $\phi_1$ and $\phi_2$ are presented in Figure 3 . The FSA for $\phi_\wedge$ (14 nodes and 72 edges) is not presented here due to space constraints.

### 6.2.2 IMPLEMENTATION DETAILS

Our policy and Q-function are represented by a neural network (3 layers each with 300, 200, 100 ReLU units). For tasks $\phi_{traverse}$ and $\phi_{interrupt}$, the input state space is 23 dimensional (22 continuous dimensional MDP state and 1 discrete dimension for the automaton state). For task $\phi_\wedge$, the state space is 24 dimensional (2 discrete dimensions for automaton states of $\phi_{traverse}$ and $\phi_{interrupt}$). We use soft Q-learning (SQL) (Haarnoja et al., 2017) to learn the optimal policies and Q-functions for $\phi_{traverse}$ and $\phi_{interrupt}$ and Theorem 2 to compose the Q-functions. The policy for task $\phi_\wedge$ is extracted from the composite Q-function by running the same algorithm on already collected experience without updating the Q-function (similar procedure as (Haarnoja et al., 2018)). After each

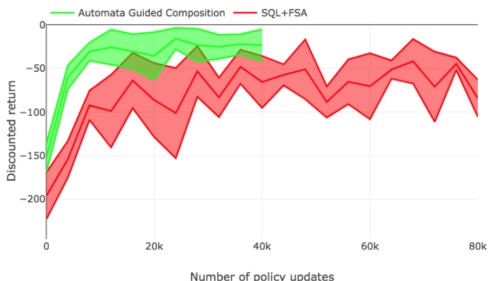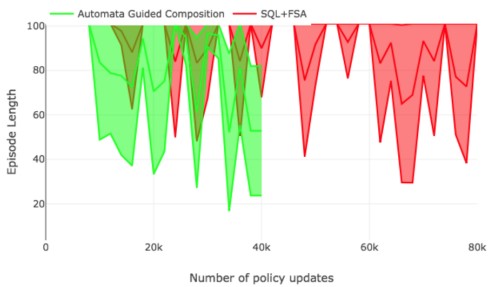

**Figure 5 : (left)** Learning curve of discounted return (discount factor 0.98). The plot shows mean and one standard deviation calculated over 5 episodes. **(right)** Mean and standard deviation of episode length (steps) averaged over 5 episodes(a smaller number means faster accomplishment of the task).

episode, the joint angles, the FSA state, the position of the plates as well as the position of the hand (represented by the yellow sphere in Figure 4 (a)) are randomly reset (within certain boundaries) to ensure generalization across different task configurations. The robot is controlled at 20 Hz. Each episode is 100 time-steps (about 5 seconds). The episode restarts if the final automaton state $q_f$ is reached. During training, we perform 100 policy and Q-function updates every 5 episodes of exploration. All of our training is performed in simulation and for this set of tasks, the policy is able to transfer to the real robot without further fine-tuning.

### 6.2.3 RESULTS AND DISCUSSION

In Figure 5 (left), we report the discounted return as a function of policy update steps for task $\phi_\wedge$. 5 evaluation episodes are collected after each set policy updates to calculate the performance statistics. As comparison, we learn the same task using SQL with FSA augmented MDP. We can see that our composition method takes less update steps to reach a policy that achieves higher returns with lower variance than the policy obtained from learning. Figure 5 (right) shows the episode length as a function of policy update (upper bound clipped at 100 steps). As mentioned in the previous section, a shorter episode length indicates faster accomplishment of the task. It can be observed that both the composition and learning method result in high variances likely due to the randomized task configuration (some plate/joint/hand configurations make the task easier to accomplish than others). However, the policy obtained from composition achieves a noticeable decrease in the average episode length.

It is important to note that the wall time for learning a policy is significantly longer than that from composition. For robotic tasks with relatively simple policy representations (feed-forward neural networks), learning time is dominated by the time used to collect experiences and the average episode length (recall that we update the policy 100 times with each 5 episodes of exploration). Since skill composition uses already collected experience, obtaining a policy can be much faster. Table 1 shows the mean training time and standard deviation (over 5 random seeds) for each task (tasks $\phi_{traverse}$, $\phi_{interrupt}$ and $\phi_\wedge$(**learned**) are trained for 80K policy updates. $\phi_\wedge$(**composed**) is trained for 40K policy updates). In general, training time is shorter for tasks with higher episodic success rate and shorter episode length. We also show the task success rate evaluated on the real robot over 20 evaluation trials. Task success is evaluated by calculating the robustness of the trajectories resulting from executing each policy. A robustness of greater than 0 evaluates to success and vice versa. $\pi_{\phi_\wedge}$(**learned**) fails to complete the task even though a convergence is reached during training. This is likely due to the large FSA of $\phi_\wedge$ with complex per-step reward ($D_\phi^q$ in Equation (7)) which makes learning difficult. Figure 4 (b) shows an evaluation run of the composed policy for task $\phi_\wedge$.

## 7 CONCLUSION

We provide a technique that takes advantage of the product of finite state automata to perform deterministic skill composition. Our method is able to synthesize optimal composite policies for $-AND-$ and $-OR-$ tasks. We provide theoretical results on our method and show its effectiveness on a grid world simulation and a real world robotic task. For future work, we will adapt our

**Table 1:** Results on Training and Evaluation Performance

| | $\phi_{traverse}$ | $\phi_{interrupt}$ | $\phi_{\wedge}$ **(learned)** | $\phi_{\wedge}$ **(composed)** |
|---|---|---|---|---|
| Mean training time/std (Hour) | 4.2/0.1 | 4.7/0.3 | 5.8/0.5 | 0.8/0.04 |
| Mean task success rate/std | 85.4%/2.3% | 80.2% / 2.5% | 3.3%/ 1% | 68.9%/ 3% |
| Mean discounted return/std | -13.2/2.3 | -10.3/3.5 | -40.6/10.5 | -20.1/8.3 |

method to the more general case of task-space transfer - given a library of optimal policies (Q-functions) that each satisfies its own specification, construct a policy that satisfies a specification that's an arbitrary (temporal) logical combination of the constituent specifications.

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
