# OpenReview forum: "Automata Guided Skill Composition"
_ICLR.cc/2019/Conference_

### Official Review · AnonReviewer1 · 2018-11-03
**More explanations are needed**

**Rating:** 5
**Confidence:** 2

**Review:**

This paper presents a way use using FSA-augmented MDPs to perform AND and OR of learned policies. This idea is motivated by the desirability of compositional policies. I find the idea compelling, but I am not sure the proposed method is a useful solution. Overall, the description of the method is difficult to follow. With more explanations (perhaps an algorithm box?), I would consider increasing my score.

The experiments demonstrate that this method can outperform SQL at skill composition. However, it is unclear how much prior knowledge is used to define the automaton. If prior knowledge is used to construct the FSA, then a missing comparison would be to first find the optimal path through the FSA and then optimize a controller to accomplish it. As the paper is not very clear, that might be the method in the paper.

Questions:
- How do you obtain the number of automaton states?
- In Figure 1, are the state transitions learned or handcoded? Are they part of the policy's action space?
- In section 3.2, you state  s_{t:t+k} |= f(s)<c ⇔ f(s_t)<c    What does s without a timestep subscript refer to? Why does this statement hold?

Can you specify more clearly what you assume known in the experiments? What is learned in the automata? In Figure 5, does SQL have access to the same information as Automata Guided Composition?

---

> ### Author Response · Authors · 2018-11-27
> **response for reviewer 1**
>
> Thank you for your comments, We try to address your questions as follows:
>
> 1. “The experiments demonstrate that this method can outperform SQL at skill composition. However, it is unclear how much prior knowledge is used to define the automaton. If prior knowledge is used to construct the FSA, then a missing comparison would be to first find the optimal path through the FSA and then optimize a controller to accomplish it. As the paper is not very clear, that might be the method in the paper. ”
>
> In this work, all prior knowledge is encoded in the scTLTL formula which effectively acts
> as a “reward function”. As mentioned in the end of Section 3.2, the automata is automatically generated from the scTLTL formula without taking additional information. Implicitly, learning with the FSA augmented MDP simultaneously finds a path in the FSA and the corresponding controller that leads the system towards the satisfying q-state. SQL and skill composition have access to the same amount of prior information.
>
> 2. “How do you obtain the number of automaton states? ”
>
> The automaton states are also automatically generated with off-the-shelf libraries. The translation from temporal logic formula to automaton is a topic in its own.
>
> 3. “In Figure 1, are the state transitions learned or handcoded? Are they part of the policy's action space?”
>
> State transitions are automatically generated with the FSA, they are not part of the action space. The states of the FSA (q states) are part of the state space and the transitions are augmented with the MDP’s transitions (definition 3).
>
> 4. “In section 3.2, you state  s_{t:t+k} |= f(s)<c ⇔ f(s_t)<c    What does s without a timestep subscript refer to? Why does this statement hold?”
>
> This statement is a definition. It says that trajectory s_{t:t+k} satisfies predicate f(s) < c if  and only if the first state of the trajectory (s_t) satisfies the predicate. For example, if the predicate is  f(s) = 2*s+1, c =5, then a trajectory {s_0=0, s_1=7, s_2=8} satisfies the predicate f(s) < c because f(s_0) < c while trajectory {s_0=7, s_1=0, s_2=-1} does not satisfy.
>
> 5. “Can you specify more clearly what you assume known in the experiments? What is learned in the automata? In Figure 5, does SQL have access to the same information as Automata Guided Composition?”
>
> Learning follows the same procedure as regular reinforcement learning. We design the scTLTL formula as task specification and we know the state and action spaces. The automata is embedded into the MDP using Definition 3. In Figure 5, SQL is used to learn a FSA augmented MDP and therefore has access to the same information.
>
> We have added a summary of our algorithm in Section 5 and updated the experiment and results section with more information and clarity.

---

### Official Review · AnonReviewer3 · 2018-11-04
**Interesting topic but little technique contribution**

**Rating:** 6
**Confidence:** 4

**Review:**

This paper mainly focuses on combining RL tasks with linear temporal logic formulas and proposed a method that helps to construct policy from learned subtasks. This method provides a structured solution for reusing learned skills (with scTLTL formulas), and can also help when new skills need to be involved in original tasks. The topic of the composition of skills is interesting. However, the joining of LTL and RL has been developed previously. The main contribution of this work is limited to the application of the previous techniques.

The proposed approach also has some limitations.
Will this method work on composing scTLTL formula with temporal operators other than disjunction and conjunction?
Can this approach deal with continuous state space and actions? This paper describes a discretization way, which, however, can introduce inaccuracies.
The design of the skills is by hand, which restricts badly its usability.
The experiments results show that the composition method does better than soft Q-learning on composing learned policies, but how it performed compared to earlier hierarchical reinforcement learning algorithms?

---

> ### Author Response · Authors · 2018-11-27
> **response for reviewer 3**
>
> Thank you for your comments. The following are our attempts to address your concerns:
>
> 1. “Will this method work on composing scTLTL formula with temporal operators other than disjunction and conjunction?”
>
> Not directly, however, if we have learned a policy for “eventually A” and a policy for “eventually B” where “A” and “B” are predicates, then it is possible to compose policies that satisfy any given scTLTL formula consisting of and only of “A”, “B”, “not A” and “not B”. This is an extension that we are working on.
>
>
> 2. “Can this approach deal with continuous state space and actions? This paper describes a discretization way, which, however, can introduce inaccuracies.”
>
> Our method is able to learn with continuous state and action spaces as is shown in the robotic experiment. The only discrete state here is the automata state which corresponds to decompositions of the high level task.
>
> 3. “The design of the skills is by hand, which restricts badly its usability.”
>
> The only hand-designed component is the scTLTL formula that specifies the task. This corresponds to the reward function that needs to be provided for most reinforcement learning algorithms.
>
> 4. “The experiments results show that the composition method does better than soft Q-learning on composing learned policies, but how it performed compared to earlier hierarchical reinforcement learning algorithms? “
>
> The FSA augmented MDP provides a natural hierarchy regardless of the RL algorithm used. Even using plain SQL results in a hierarchical policy. The reason we did not compare our method with other RL algorithms on a regular MDP is that it is difficult to specify a complex task using a non-temporal logic reward function. In our experience, if enough effort is put into reward design, we will end up with something very similar to the robustness of the original scTLTL formula, anything less will result in a faulty reward that makes the comparison less meaningful. Again, the focus of this work is more on the effective composition of learned skills and less on actually learning a skill.
>
> We have incorporated a summary of our algorithm in Section 5 and also updated the experiment and results section with more information.

---

### Official Review · AnonReviewer4 · 2018-11-11

**Rating:** 5
**Confidence:** 2

**Review:**

This work proposed using temporal logic formulas to augment RL learning via the composition of previously learned skills. This work was very difficult to follow, so it is somewhat unclear what were the main contributions (since much of this seems to be covered by other works as referenced within the paper and as related to similar unreferenced works below). Moreover, regarding the experiments, many things were unclear (some of the issues are outlined below). While the overall idea of using logic in this way to help with skill composition is interesting and exciting, I believe several things must be addressed with this work. This includes: situating this work more clearly against existing similar works which use logic in this way, clearly defining the novel contributions of this work as compared to those and others, overall making the methodology more clear and specific (including experimental methodology), and comparing/contrasting against (or at least discussing differences with) methods with similar motivations (e.g., HRL multi-task learning, meta-learning) to emphasize the need/importance of this work — I am aware that at least 1 HRL work is mentioned, but this work is not really contrasted against it to help situate it.

Questions/Concerns about Experiments:

+ Does Figure 5 show the averaged return over 5 runs, sum of discounted rewards averaged over 5 episodes per update step, or 5 episodes, each from a separate run averaged together? It is a bit unclear especially because the main text and the figure caption slightly differ. Also, average discounted return is somewhat different than average return,  suggest updating the label to be clear also with the discount factor used.
+ What were the standard deviations for this across experiments? Even with averaging it seems that these runs are very high variance, would be good to understand what variance bounds to expect if using this method.
+ Why were average discounted returns reported in Figure 5 and not in Table 1?
+  What were the standard deviations on success rate and training time? Also what about sample complexity?
+ To my understanding the benefit here is reusability of learned skills via the automata methods described here. It would have made sense to compare against other HRL or multi-task learning methods in addition to just SQL or learning from scratch. For example how would MAML compare to this?
+ It is also unclear whether the presented results in Table 1 and Figure 5 are on the real robot or in simulation. The main text says, “All of our training is performed in simulation and the policy is able to transfer to the real robot without further fine-tuning.” So does this mean that Figure 5 is simulated results and Table 1 is on the real robot?



Citations that should likely be made:

+ Giuseppe, Luca Iocchi, Marco Favorito, and Fabio Patrizi. "Reinforcement Learning for LTLf/LDLf Goals." arXiv preprint arXiv:1807.06333 (2018).
+ Camacho, Alberto, Oscar Chen, Scott Sanner, and Sheila A. McIlraith. "Decision-making with non-markovian rewards: From LTL to automata-based reward shaping."  In Proceedings of the Multi-disciplinary Conference on Reinforcement Learning and Decision Making (RLDM), pp. 279-283. 2017.
+ Camacho, Alberto, Oscar Chen, Scott Sanner, and Sheila A. McIlraith. "Non-Markovian Rewards Expressed in LTL: Guiding Search Via Reward Shaping." In Proceedings of the Tenth International Symposium on Combinatorial Search (SoCS), pp. 159-160. 2017.


Typos/Suggested grammar edits:

“Skills learned through (deep) reinforcement learning often generalizes poorly across tasks and re-training is necessary when presented with a new task.” —> Often generalize poorly

“We present a framework that combines techniques in formal methods with reinforcement learning (RL) that allows for convenient specification of complex temporal dependent tasks with logical expressions and construction of new skills from existing ones with no additional exploration.” —> Sentence kind of difficult to parse and is a run-on

“Policies learned using reinforcement learning aim to maximize the given reward function and is often difficult to transfer to other problem domains.” —> ..and are often..

“by authors of (Todorov, 2009) and (Da Silva et al., 2009)” —> by Todorov (2009) and Da Silva et al. (2009) Also several other places where you can use \citet instead of \cite

---

> ### Author Response · Authors · 2018-11-27
> **response for reviewer 4**
>
> Thank you for your comments and providing additional references. We try to address your questions as follows:
>
> 1. “situating this work more clearly against existing similar works which use logic in this way, ..., and comparing/contrasting against (or at least discussing differences with) methods with similar motivations (e.g., HRL multi-task learning, meta-learning) to emphasize the need/importance of this work”
>
> To the best of our knowledge, the presented work is the first to use techniques in formal methods to simultaneously address optimal -AND- and -OR- task compositions and demonstrate the process in tasks with continuous state and action spaces. We make the distinction between skill composition and multi-task learning/meta-learning (such as MAML) where the latter often requires a predefined set of tasks/task distributions to learn and generalize from, whereas the focus of the former is to construct new policies from a library of already learned policies that achieve new tasks (often some combination of the constituent tasks) with little to no additional constraints on task distribution at learning time. Our focus here is on task composition and therefore did not compare with multi-task / meta learning methods. HRL is also not the focus here, it so happens that incorporating FSA into the MDP gives the resulting policy a hierarchical representation. Therefore, we chose mainly to contrast against other skill composition methods in Section 2. We have made this more clear in the updated paper.
>
>  2. "Citations that should likely be made ..."
>
> The set of provided references aim to solve the non-Markovian reward decision process (NMRD) using temporal logic and automaton. The idea is similar to that of the FSA augmented MDP that we adopted with some differences (such as the requirement to manually define a set of rewards in additional to the logic specification, separation of state features and temporal goals, etc). However, the comparison is mainly between the above references and the FSA augmented MDP (Li et al., 2018) which is not the contribution of our work.
>
> 3. “Does Figure 5 show the averaged return over 5 runs, sum of discounted rewards averaged over 5 episodes per update step ...”
>
> The original Figure 5 shows the undiscounted episodic return (sum of undiscounted rewards over one episode) averaged over 5 evaluation episodes (without updating the policy in between). We have updated this result to be discounted return with standard deviations.
>
> 4. “What were the standard deviations for this across experiments? Even with averaging it seems that these runs are very high variance, would be good to understand what variance bounds to expect if using this method.”
>
> We have included the standard deviation in the learning curve. To our current understanding, the variance comes from two sources. The first is randomization of the environment configurations - some configurations make the task considerably easier to accomplish than others. The second is randomization over the automaton states at initialization. Some q states as easier to learn than others (for example $q_2$ compared to $q_1$ in Figure 4b). At each initialization, if a difficult q state is on the agent’s path of reaching $q_f$, the agent may get stuck in that state receiving a low episodic return whereas in other episodes the agent may not have to deal with this state at all.
>
> 5. “Why were average discounted returns reported in Figure 5 and not in Table 1?”
>
> Originally, Table 1 aims to report the performance of the learned policies in terms of task success rate whereas Figure 5 reports learning progress in terms of returns. We have updated Table 1 to include the average discounted returns.
>
> 6. “What were the standard deviations on success rate and training time? Also what about sample complexity?”
>
> We have added the standard deviations to Table 1. We don't currently have a quantitative analysis on sample complexity other than the learning curves. Hopefully we will perform such analysis in the future.
>
> 7. “It is also unclear whether the presented results in Table 1 and Figure 5 are on the real robot or in simulation. The main text says, “All of our training is performed in simulation and the policy is able to transfer to the real robot without further fine-tuning.” So does this mean that Figure 5 is simulated results and Table 1 is on the real robot?”
>
> This is correct, training is in simulation and evaluation is on the real robot. We have modified the text to make this clear in the paper.
>
> Thank you also for catching the typos and suggesting grammar edits, those have been incorporated in the updated paper. We have also updated the experiment and results section.

---

> > ### Comment · AnonReviewer4 · 2018-12-09
> > **Appreciate Improvements**
> >
> > I'd like to say that I appreciate the improvements to the paper and have updated my previous rating accordingly. I'm still not totally convinced that the other methods I mentioned aren't relevant and also now I have some
> > mild concerns about the high variance in the std reported, making it difficult to assess real performance gains are real or not (right side of figure 5). I would also liked to have seen the additional analysis mentioned in this post.

---

> > > ### Author Response · Authors · 2018-12-19
> > > **Thanks for the followup**
> > >
> > > Thank you for the additional comments and adjustment to the score. We acknowledge that comparison with a good number of state-of-the-art methods would better situate our work in the field. Our work presented here is a combination of both reward engineering (using TL) and skill composition, along with the hierarchical policy structure that arises naturally with the framework. It is difficult to find other work with a similar combination. Therefore, an elaborate and fair comparison with other methods would be a contribution in it self which we will consider in the future. As for the high variance on the right side of figure 5, it mostly depends on how the task is initialized at each episode (initialization is random). Some initialization makes it easier for the task to be accomplished than others. As long as the episode length is consistently below the max value, the agent is always able to complete the task. As the reviewer mentioned, a more thorough analysis will be helpful which we will try to incorporate in future work.

---

### Official Review · AnonReviewer2 · 2018-11-27
**Nice paper that combines RL and constraints expressed by logical formulas**

**Rating:** 7
**Confidence:** 3

**Review:**

The contribution of the paper is to set up an automaton from scTLTL formulas, then corresponding MDP that satisfies the formulas is obtained by augmenting the state space with the automaton state and zeroing out transitions that do not satisfy the formula. This approach seems really useful for establishing safety properties or ensuring that constraints are satisfied, and it is a really nice algorithmic framework. The RL algorithm for solving the problem is entropy-regularized MDPs. The approach “stitches” policies using AND and OR operators, obtaining the overall optimal policy over the aggregate. Proofs just follow definitions, so they are straightforward, but I think this is a quality. The approach is quite appealing because it provides composition automatically. The paper is very well written.  The main problem I see with the work is that composition can explode the number of states in the new automaton and hence the new MDP. It would be interesting in future work to do “soft” ruling out of transitions rather than the "hard" approach used in the paper. The manipulation task provided is quite appealing, as the robot arm is of high dimensionality but the FSAs obtainedare discrete. Overall, the paper provides a very good contribution.

Small comments:
Equation equation in Def 3 also proof of Theorem 2
In section,  -> In this section
are it has -> and it has

---

> ### Author Response · Authors · 2018-11-27
> **Response for reviewer 2**
>
> Thank you for your comments.  The dimensional explosion of automaton states when composing many policies and its effect on composition is an interesting and practical problem worth looking into. Thank you also for catching the typos, we have incorporated the modifications in the updated paper.

---

### Meta-Review · Area_Chair1 · 2018-12-14
**Interesting combination of temporal logic for constructing new RL policies, presentation should be clearer**

**Confidence:** 3
**Recommendation:** Reject

**Metareview:**

The authors present an interesting approach for combining finite state automata to compose new policies using temporal logic. The reviewers found this contribution interesting but had several questions that suggests that the current paper presentation could be significantly clarified and situated with respect to other literature. Given the strong pool of papers, this paper was borderline and the authors are encouraged to revise their paper to address the reviewers’ feedback.